# Characterization and Modeling of Free Volume and Ionic Conduction in Multiblock Copolymer Proton Exchange Membranes

**DOI:** 10.3390/polym14091688

**Published:** 2022-04-21

**Authors:** Mahmoud Mohammed Gomaa, Arturo Sánchez-Ramos, Nieves Ureña, María Teresa Pérez-Prior, Belen Levenfeld, Pablo A. García-Salaberri, Mohamed Rabeh Mohamed Elsharkawy

**Affiliations:** 1Physics Department, Faculty of Science, Minia University, Minia P.O. Box 61519, Egypt; mahmoud_gomaa19@mu.edu.eg (M.M.G.); mrmelsharkawy@mu.edu.eg (M.R.M.E.); 2Department of Thermal and Fluids Engineering, Universidad Carlos III de Madrid, 28911 Leganes, Spain; artsanch@pa.uc3m.es; 3Department of Materials Science and Engineering and Chemical Engineering, Universidad Carlos III de Madrid, 28911 Leganes, Spain; murena@ing.uc3m.es (N.U.); maperezp@ing.uc3m.es (M.T.P.-P.); bll@ing.uc3m.es (B.L.)

**Keywords:** PALS, electrochemical impedance spectroscopy, modeling, ionic conductivity, free volume, SPES, proton exchange membrane

## Abstract

Free volume plays a key role on transport in proton exchange membranes (PEMs), including ionic conduction, species permeation, and diffusion. Positron annihilation lifetime spectroscopy and electrochemical impedance spectroscopy are used to characterize the pore size distribution and ionic conductivity of synthesized PEMs from polysulfone/polyphenylsulfone multiblock copolymers with different degrees of sulfonation (SPES). The experimental data are combined with a bundle-of-tubes model at the cluster-network scale to examine water uptake and proton conduction. The results show that the free pore size changes little with temperature in agreement with the good thermo-mechanical properties of SPES. However, the free volume is significantly lower than that of Nafion^®^, leading to lower ionic conductivity. This is explained by the reduction of the bulk space available for proton transfer where the activation free energy is lower, as well as an increase in the tortuosity of the ionic network.

## 1. Introduction

Proton exchange membrane fuel cells (PEMFCs) are receiving considerable attention for stationary and mobile applications because of their attractiveness as efficient and eco-friendly energy converters. Areas of interest include unmanned aerial vehicles (UAVs), transportation sector (heavy duty trucks, light duty vehicles, trains, submarines, etc.) and forklifts, among other devices. The main advantage of PEMFCs compared to batteries is their increased operational time, such as extended flight time of UAVs or extended range of trucks. Another advantage is no emission of air pollutants, such as forklifts in enclosed facilities and widespread use of light duty vehicles [1].

A key component of PEMFCs is the polymer membrane (PEM), which plays a key role in ohmic losses and water management [2,3]. Nafion^®^ is widely used as a polymer electrolyte in PEMFCs. It is composed of a hydrophobic polytetrafluoroethylene (PTFE) backbone (good chemical stability) and hydrophilic sulfonic acid groups (high water uptake). Upon hydration, there is significant phase separation between hydrophobic and hydrophilic domains, thus providing well-defined channels for proton conduction. However, the special structure of Nafion^®^ results in high costs due to its complex synthesis procedure [4,5,6]. The main challenge for obtaining alternative electrolytes is the improvement of the stability under operational conditions (i.e., temperature and relative humidity) of the current commercial electrolytes [7,8]. This challenge necessitates the search for other ionomers based on different polymeric backbones such as poly(2,6-dimethyl-1,4-phenylene oxide) (PPO), poly(arylene ether ketone) (PAEK), polystyrene (PS), polybenzimidazole (PBI), polysulfone (PSU), or poly(phenylsulfone) (PPSU) [9,10]. A good PEM should have the following properties: (i) high proton conductivity (>10^−1^ S cm^−1^) at operating conditions, associated with its ion exchange and water sorption capacities and morphology of ionic channels, (ii) low permeation of undesired crossover species, (iii) good thermal and chemical stability, and (iv) easy and low-cost synthesis and scale-up [10]. Recently, multiblock copolymers have been proposed as promising electrolytes for PEMFCs (see, e.g., [11,12,13]). They are an emerging class of synthetic polymers that exhibit different macromolecular structures and behavior to those of homopolymers or di/triblock copolymers, with the potential to outperform common perfluorosulfonic acid (PFSA) PEMs (e.g., Nafion^®^). In this context, there are hardly any published works on copolymers of styrene and polysulfone and even less for PEMFCs [14]. As an example, polystyrene-polysulfone-polystyrene triblockcopolymers were synthesized as membranes for fuel cell applications [15]. However, there are some published works describing the preparation and characterization of various sulfonated poly(arylene ether sulfone)-based copolymers due to their good properties [16,17,18]. Specifically, SPSU and SPPSU are well known for the preparation of membranes due to their excellent physicochemical properties, including good thermal stability and chemical resistance, sufficient mechanical strength, and good processability [19]. In addition, specific advantages of using SPES membranes must be highlighted. The preparation of these copolymers can be carried out starting from high purity, commercial and cheap monomers, allowing the development of competitive membranes with very high performance from an electrochemical and mechanical point of view.

Multiple PEM characteristics that affect cell performance, such as water uptake, mechanical properties and gas permeability, are all strongly correlated with the structure of free volume holes (i.e., gaps between entangled polymer chains). The analysis of the size and structure of ionic channels is crucial to determining optimal preparation routes and operating conditions for maximum performance. The best technique for this task is positron annihilation lifetime spectroscopy (PALS), which has been successfully used to study the free volume of many materials with high accuracy [20,21,22,23,24,25,26].

PALS is a very sensitive and non-destructive technique for the study of free volume holes and their distribution in a range lower than nano scale. The basic idea of PALS system is to count the time difference between the positron entry into the PEM and its death after the interaction with the surrounding electrons of the material. During the thermalization of positrons in the PEM, several processes may occur, depending on the structure of the material. One possibility is that each positron is annihilated with the free electrons and emits two gamma photons with a fixed energy of 511 keV for each photon. Alternatively, each positron can pick up one of the atomic electron-forming positronium (Ps atom), which will also face the same end as it will also annihilate but with a different lifetime. There are two types of Ps atoms, according to the spin direction of the electron and positron. The parapositronium (*p*-Ps) has an anti-parallel spin and a lifetime of about 0.125 ns, while the ortho-positronium (*o*-Ps) has a parallel spin and lives for longer inside the material (1–10 ns). When examining polymers, the most important value is the ortho-positronium (*o*-Ps) lifetime, which correlates with the free volume radius.

In this work, multiblock copolymer PEMs based on sulfonated PSU and PPSU are examined. This “hydrophilic-block-hydrophilic” multiblock copolymer was chosen because: (i) it can be manufactured on a large scale, (ii) it shows good water sorption capacity and mechanical strength, and (iii) preliminary MEA tests were very promising [12,13]. The use of “hydrophobic-block-hydrophilic” copolymers by concentrating charge in one block and using the other as a backbone is also an interesting approach. However, the tailored design and synthesis of “hydrophilic-block-hydrophilic” copolymers is easier from an industrial point of view. To the best of our knowledge, few reports combine experimental and numerical studies of alternative structures to understand how proton conduction works. The aim of this work is to examine experimentally and numerically the pore size distribution of ionic channels and the proton conductivity of sulfonated PSU and PPSU multiblock copolymers. Both the effect of relative humidity and temperature are assessed.

The organization of the paper is as follows. In Section 2, the synthesis procedure is presented. In Section 3, the methods used for PEM characterization are described: water uptake, swelling ratio, water volume fraction, pore size distribution and proton conductivity. In Section 4, the bundle-of-tubes model used for the analysis of the PEMs is presented. In Section 5, the results are discussed in terms of pore size distribution, water volume fraction and proton conductivity. Finally, the conclusions are given in Section 6.

## 2. Materials

The synthesis pathway of the sulfonated PSU/PPSU copolymer membranes was previously described in [12] (see Figure 1). Copolymers were obtained via polycondensation using a “one-pot two-step synthesis” [27]. Purified and dried reagents were dissolved in *N*,*N*-dimethylacetamide (DMAc, Acros Organic, Geel, Belgium). The reaction to obtain the PPSU block was maintained at 120 ℃ for 18 h in a flask under inert atmosphere. Once the reaction was over, the reagents dissolved in DMAc for the PSU block were added in the same flask and maintained at 120 ℃ for 18 h more. Toluene (Sigma-Aldrich, Sant Louis, MO, USA) was used as an azeotropic agent. The copolymer was precipitated in a 1 M HCl solution and dried under vacuum at 60 ℃ for 48 h. The synthesized poly(ether sulfone)s (PES) multiblock copolymers show similar PSU/PPSU number ratios of ∼1:1, well-controlled molecular weights of each block (5000 g mol^−1^) and relatively small polydispersity [12].

The sulfonation reaction was carried out according to Chao et al. [28]. Synthesized PES copolymer was dissolved in dry 1,2-dichloroethane (DCE, Sigma-Aldrich, Sant Louis, MO, USA) under inert atmosphere at ambient temperature. Subsequently, the sulfonating agent trimethylsilyl chlorosulfonate (TMSCS, Sigma-Aldrich, Sant Louis, MO, USA), previously dissolved in DCE (1:3, 1:6 and 1:9 PSU:TMSCS molar ratio), was added dropwise and maintained for 24 h. The polymers (SPES-Na) were precipitated in a 0.1 M solution of sodium hydroxide and dried under vacuum at 60 ℃. The sulfonation reaction of PES was performed using TMSCS due to the lower degradation of polymer chains. Three different degrees of sulfonation (DS) were prepared according to the sulfonating agent added. The DS of copolymers was calculated using the individual DS of each block according to [13]
(1)$$DS=\frac{nDS_{PSU}+mDS_{PPSU}}{n+m}$$
where n and m are the number of structural units (molecules) of the sulfonated *PSU* and *PPSU* blocks. Hereafter, the synthesized copolymer PEMs are abbreviated according to their DS as SPES 1 (low), SPES 2 (medium-high), and SPES 3 (high). See Table 1.

Ion exchange capacity (IEC) was determined by both acid-base titration in aqueous solution and titration in an organic solvent (see [12] for further details). Density was measured after drying at 60 ℃ under vacuum.

## 3. Experimental

Ionomers were dissolved in DMAc (5 wt%) and casted onto a petri glass and dried under vacuum for over 48 h. The resulting thickness was 50 µm. Finally, SPES-Na was immersed in a 1 M HCl solution at 60 ℃ for 24 h to obtain the proton form (SPES-H).

### 3.1. Water Uptake, Swelling Ratio and Water Volume Fraction

The water uptake, WU, was determined as a function of relative humidity and temperature (T=80 °C, RH=0.1−0.8, and RH=0.8, T=30−80 °C according to
(2)$$WU=\frac{M_{wet}-M_{dry}}{M_{dry}}$$
where $M_{wet}$ and $M_{dry}$ are the mass of the humidified and dry PEMs, respectively. The measured WU data are listed in Table 2.

The volumetric swelling ratio, SW, and water volume fraction, $\phi_{v}$, were determined from the WU, using the following expressions
(3)$$SW=\frac{V_{wet}-V_{dry}}{V_{dry}}=\left(1+WU\right)\frac{\rho_{dry}}{\rho_{wet}}-1\;\phi_{v}=\frac{V_{w}}{V_{wet}}=\frac{WU/\rho_{w}}{1/\rho_{dry}+WU/\rho_{w}}$$

Here, $V_{wet},\rho_{wet}$ and $V_{dry},\rho_{dry}$ are the volume, density of the humidified and dry PEMs, respectively, and $\rho_{w}$ is the water density. According to the rules of mixtures, $\rho_{wet}$ is given by
(4)$$\rho_{wet}=\frac{M_{wet}}{V_{wet}}=\phi_{v}\rho_{w}+\left(1-\phi_{v}\right)\rho_{dry}\;$$
where $M_{wet}=M_{dry}+M_{w}$ is the mass of the humidified PEM.

### 3.2. Pore Size Distribution

As shown in Figure 2, PALS measurements were performed with a fast-fast coincidence system (260 ps resolution) under two different operating conditions: (i) constant temperature and variable RH (T=30 °C, RH%=30−80), and (ii) constant RH and variable T (RH%=0 (dry), T=30−90 °C). ^22^NaCl radioactive material with an activity of approx. 0.28 MBq was used as a positron source and was enveloped between 7 μm Kapton foil. The membranes were cut into 1 × 1 cm² pieces, and a stack of the membrane layers were made to enclose the source on both sides. The membrane-positron source-membrane sandwich was then placed in a vacuum chamber with a humidity control system with a resolution of ±1% RH. The PALSfit3 software (PALSfit3, Version 3.104, Jens V. Olsen, Peter Kirkegaard and Morten Eldrup Technical University of Denmark) was used to analyze the spectra after collecting approximately 2 million counts, taking about 2 h for each spectrum (Appendix A about the PALS measurements can be found in Appendix A). A high purity Si sample was used to determine the source correction, which was found to be 10.2%. These values were then removed from the collected spectra. The *o*-Ps lifetime distribution was obtained by analyzing the PAL spectra, assuming a log-normal distribution [29].

In polymers, Ps atom is usually generated due to holes or open spaces between polymer chains. The resulting PALS spectrum contains three-lifetime components ($\tau_{1}$, $\tau_{2}$ and $\tau_{3}$) with relative intensities ($I_{1}$, $I_{2}$ and $I_{3}$) and in some cases contains more. The first lifetime ($\tau_{1}$) refers to *p*-Ps, the second lifetime ($\tau_{2}$) to the free positron and the third lifetime ($\tau_{3}$) to the longest lifetime, which is associated with *o*-Ps pick-off annihilation. The third lifetime is the most important component since it can be correlated to the mean radius of holes, R, according to the semi-empirical equation of Tao-Eldrup (assuming that Positronium is in a spherical potential with an infinite potential barrier of radius R and an electron layer ΔR) [30]
(5)$$\tau_{o-\mathrm{Ps}}=0.5\left\{1-\frac{R}{R_{o}}+\frac{1}{2\pi \;}\sin\left(\frac{2\pi R}{R_{o}}\right)\;\right\}\;$$
where $R_{o}=R+\Delta R$ and ΔR=1.656 Å is estimated from a material with a known free volume and represents the thickness of the homogeneous electron layer in which positron annihilates.

From Equation (5), the characteristic free volume radius R and the average *o*-Ps hole volume ($V_{\mathrm{Ps}}$) can be determined as follows
(6)$$V_{\mathrm{Ps}}=\frac{4}{3}\;\pi R^{3}\;$$

A positron lifetime spectrum can be represented by
(7)$$S\left(t\right)=N_{t}R\left(t\right)C\left(t\right)$$
where S(t) is the collected spectrum, $N_{t}$ is the total count, R(t) is the resolution function, and
(8)$$C\left(t\right)=\overset{n}{\underset{i=1}{\sum }}\alpha_{i}\lambda_{i}e^{-\lambda_{i}t}d\lambda \;$$
where n is the number of lifetime components, $\alpha_{i}$ is the relative intensity of the *i*th component, $\lambda_{i}$ is the annihilation rate (inverse of lifetime) and t is time. Equation (8) represents a decomposition into discrete lifetimes. If we assume that each λ has a distribution, Equation (8) becomes
(9)$$C\left(t\right)=\overset{\infty }{\underset{0}{\int }}\alpha_{i\;}\lambda_{i}e^{-\lambda_{i}t}\;d\lambda \;$$

The fractional free volume, $F_{v}$, is the ratio between the free volume to the total volume of the material and can be calculated by
(10)$$F_{v}=CV_{h}I_{3}\;$$
where C is a constant equal to 0.018 nm^−3^, $V_{h}$ is the mean free volume and $I_{3}\%$ is the *o*-Ps intensity. Using the correlation between $\tau_{3}$ and the radius R, Equation (5), the void radius probability density function f(R) can be written as [31]
(11)$$f\left(R\right)=2\Delta R\left[\cos\left(\frac{2\pi R}{R+\Delta R}\right)-1\;\right]\times \frac{\alpha \left(\lambda \right)}{\left[{\left(R+\Delta R\right)}^{2}\right]}\;$$

Equation (11) can be modified to obtain the volume probability density function [32]
(12)$$g\left(V\right)=2\Delta R\left[\cos\left(\frac{2\pi R}{R+\Delta R}\right)-1\;\right]\times \frac{\alpha \left(\lambda \right)}{\left[4\pi R^{2}{\left(R+\Delta R\right)}^{2}\right]}\;$$

### 3.3. Proton Conductivity

Proton conductivity was measured at T=80 °C (RH=0.1−0.8) and RH=0.8 (T=30−80 °C) by means of electrochemical impedance spectroscopy (EIS), using a Hewlett Packard 4192A impedance analyzer (Yokogawa-Hewlett Packard LTD., Tokyo, Japan). The experiments were carried out in a conductivity cell composed of two gold electrodes separated by a PEM in the frequency range between $10^{-1}$ and $10^{6}$ Hz (0.01 V voltage amplitude). A Vösch 4018 climate chamber was used to control temperature and relative humidity. The membrane resistance was determined by the frequency intercept with the real axis in the Nyquist plot. An example of the Nyquist plots is given in Figure 3, together with the equivalent circuit used to fit the experimental data. The impedance of the in-house device was measured by shorting the electrodes and the value (0.2 Ω) of the resistance (*R_c_*) was introduced in the fit equation. The membrane impedance is composed of the bulk resistance, *R_b_*, in parallel with the bulk membrane capacitance, *C_b_*. The behavior at electrode/membrane interfaces is merely capacitive (represented in the circuit used to fit by *CPEdl* [33]) and appears at low frequency in the impedance spectra.

The proton conductivity was determined from the measured membrane resistance, $R_{m}$, according to the expression
(13)$$\kappa =\frac{\delta }{R_{m}A}\;$$
where δ and A are the thickness and active area, respectively. The experimental data obtained from EIS measurements were analyzed using the Z-View analysis impedance software (version 2.9 c, Scribner Associates, Inc., Southern Pines, NC, USA) and are listed in Table 2 together with the WU data.

## 4. Modeling

Proton conduction was analyzed using a bundle-of-tubes model implemented in MATLAB (Natick, MA, USA). As shown in Figure 4, the free volume was divided into three types of pore bodies: (i) hydrated sulfonated sites, (ii) dry sulfonated sites, and (iii) dry non-sulfonated sites [13]. The radius of the tubes, r, was determined based on the PALS measurements at different RH (RH=0−0.8) and temperature (T=30−90 °C) by non-linear fitting of experimental data to log-normal distributions [34]
(14)$$PSD\left(r\right)=\frac{1}{r\sigma \sqrt{2\pi }}\exp\;\left[\frac{-{\left(\ln\;r-\ln\;r_{c}\right)}^{2}}{2\sigma^{2}}\right]\;$$
where $r_{c}$ is the characteristic pore radius and σ is the standard deviation. The moments of the log-normal distribution are provided in Appendix B.

The relationships of $r_{c}$ and σ with RH and T from PALS at T=30 °C (RH=0−0.8) and RH=0 (T=30−90 °C) were extended on the full RH−T plane by bilinear interpolation, given the rather linear variations found in the experimental data (information about the interpolated distributions can be found in Appendix A). The interpolated values allowed us to determine the pore size distributions of the PEMs and examine the proton conductivity at RH=0.8 (T=30−80 °C). The variation of $\chi =r_{c},\;\sigma$ as a function of RH and T is given by the following expression in terms of the weight functions $w_{ij}\left(\mathrm{RH},T\right)$
(15)$$P\chi \left(\mathrm{RH},T\right)=w_{11}\left(\mathrm{RH},T\right)\chi_{11}+w_{12}\left(\mathrm{RH},T\right)\chi_{12}+w_{21}\left(\mathrm{RH},T\right)\chi_{21}+w_{22}\left(\mathrm{RH},T\right)\chi_{22}\;$$where
(16)$$w_{11}\left(\mathrm{RH},T\right)=\frac{\left({\mathrm{RH}}_{2}-\mathrm{RH}\right)\left(T_{2}-T\right)}{\left({\mathrm{RH}}_{2}-{\mathrm{RH}}_{1}\right)\left(T_{2}-T_{1}\right)},\;w_{12}\left(\mathrm{RH},T\right)=\frac{\left({\mathrm{RH}}_{2}-\mathrm{RH}\right)\left(T-T_{1}\right)}{\left({\mathrm{RH}}_{2}-{\mathrm{RH}}_{1}\right)\left(T_{2}-T_{1}\right)}\;w_{21}\left(\mathrm{RH},T\right)=\frac{\left(\mathrm{RH}-{\mathrm{RH}}_{1}\right)\left(T_{2}-T\right)}{\left({\mathrm{RH}}_{2}-{\mathrm{RH}}_{1}\right)\left(T_{2}-T_{1}\right)},\;w_{22}\left(\mathrm{RH},T\right)=\frac{\left(\mathrm{RH}-{\mathrm{RH}}_{1}\right)\left(T-T_{1}\right)}{\left({\mathrm{RH}}_{2}-{\mathrm{RH}}_{1}\right)\left(T_{2}-T_{1}\right)}$$

In the above expression, the reference points 1 and 2 on the RH−T plane are equal to ${\mathrm{RH}}_{1,2}=0,\;0.8$ and $T_{1,2}=30,\;90\;{}^{\circ}C$. The reference values $\chi_{ij}$ are listed in Table 3.

### 4.1. Assumptions

The model is based on the following simplifying assumptions:
Copolymer PEMs are macroscopically homogeneous.Number ratio of sulfonated tubes is equal to DS.Pore volume of tubes is equal to free volume.Non-sulfonated tubes are not hydrated and therefore are non-conductive. Electroneutrality holds in hydrated tubes.Surface charge density of sulfonic groups ${\mathrm{SO}}_{3}^{-}$ is homogeneous without distinction between copolymer blocks. That is, the average spacing of ${\mathrm{SO}}_{3}^{-}$ groups over the wet ionic network does not change significantly.Convection is negligible.

### 4.2. Volume Fraction of Water and Hydrated Tubes

The volume fraction of water in the copolymer PEMs is given by [13]
(17)$$\phi_{v}=\phi_{w}\varepsilon_{w}=DS\phi_{wr}\varepsilon_{w}\;$$
where $\phi_{w}$ is the volume fraction of hydrated tubes, $\phi_{wr}=\phi_{w}/DS$ is the relative volume fraction of hydrated tubes (i.e., the fraction of sulfonated tubes that are hydrated), and $\varepsilon_{w}$ is the average water volume fraction (i.e., water-filled porosity) in each representative cube of length $d_{c}$.

The relative volume fraction of hydrated tubes, $\phi_{wr}$, depends on humidification, so that $\phi_{wr}=0$ under dry conditions (RH=0) and $\phi_{wr}=1$ under fully humidified conditions (RH=1) [13]. A cubic relationship is used, as typically considered to correlate water uptake as a function of RH [11]
(18)$$\phi_{wr}\left(\mathrm{RH}\right)=a_{1}{\mathrm{RH}}^{3}+a_{2}{\mathrm{RH}}^{2}+a_{3}\mathrm{RH};\;\;a_{1}+a_{2}+a_{3}=1\;$$

The dimensionless coefficients $a_{i}$ are listed in Table 3.

The average water-filled porosity, $\varepsilon_{w}$, is determined according to the free volume. Tortuous tubes of radius r and characteristic length, $L_{c}=\left(4/3\right)r$, are assumed to make the pore volume of the tubes equal to that of spherical cavities from PALS measurements, i.e.,
(19)$$\varepsilon_{w}=\frac{V_{w,h}}{V_{h}}\;$$
where $V_{w,h}$ and $V_{h}$ are the average volumes of water and hydrated sites, respectively, as given by the following expression
(20)$$V_{w,h}=\frac{4}{3}\pi \frac{I_{3}\left(r_{a},r_{b}\right)}{I_{0}\left(r_{a},r_{b}\right)};\;V_{h}=d_{c}^{3}\;$$

Here, $I_{3}\left(r_{a},r_{b}\right)$ and $I_{0}\left(r_{a},r_{b}\right)$ are the third- and zero-order moments of PSD(r) in the interval $\left[r_{a},r_{b}\right]$ (see Appendix B), and $d_{c}=d_{o}{\left(1+SW\right)}^{1/3}$, being $d_{o}$ the characteristic spacing between ionic tubes under dry conditions.

The range of hydrated tube radii, $\left[r_{a},r_{b}\right]$, increases around the median, $\tilde{r}$, depending on $\phi_{w}\left(\mathrm{RH}\right)=DS\phi_{wr}\left(\mathrm{RH}\right)$, so that the cumulative probability of finding a hydrated tube with a radius below and above $\tilde{r}$ is equal to $\phi_{w}/2$
(21)$$I_{0}\left(r_{a},\;\tilde{r}\right)=\frac{\phi_{w}}{2}\Rightarrow r_{a}=\tilde{r}\exp\left[\sqrt{2}\sigma (\phi_{w})\;\right]\;I_{0}\left(\tilde{r},r_{b}\right)=\frac{\phi_{w}}{2}\Rightarrow r_{b}=\tilde{r}\exp\left[-\sqrt{2}\sigma (\phi_{w})\;\right]$$

All tubes are hydrated when the pore space is fully sulfonated (DS=1) and filled with water ($\phi_{wr}=1$), whereas no tubes are hydrated under dry conditions [13].

### 4.3. Proton Conductivity at the Cluster Scale

According to the Nernst–Planck equation, in the absence of a pressure gradient, the flux of protons, $N_{H^{+}}$, is composed of migration and electro-convection, since the diffusive flux vanishes due to the electroneutrality condition ($dC_{H^{+}}/dx=0$) [35]. Moreover, the electro-kinetic velocity induced by the electrostatic field in electric double layers can be neglected in small pores of vapor-equilibrated copolymer PEMs, $r^{avg}∼10^{-1}\;\mathrm{nm}$. Therefore, the current density, I, is given by
(22)$$N_{H^{+}}=\frac{FD_{H^{+},w}\left(T\right)\;}{RT}C_{H^{+}}^{f}\frac{d\varphi }{dx}\Rightarrow I=\frac{F^{2}D_{H^{+},w}\left(T\right)\;}{RT}C_{H^{+}}^{f}\frac{d\varphi }{dx}\;$$

The effective proton conductivity at the cluster scale is equal to
(23)$$\kappa_{c}=\frac{I}{d\varphi /dx}=\frac{F^{2}D_{H^{+},w}\left(T\right)\;}{RT}C_{H^{+}}^{f}\;$$
where x is the coordinate across the PEM thickness, $D_{H^{+},w}\left(T\right)$ is the diffusivity of protons in liquid water and $C_{H^{+}}^{f}$ is the average proton concentration in the fluid phase. The dependence of $D_{H^{+},w}$ with T is modeled by the Stokes-Einstein law [36]
(24)$$D_{H^{+},w}\frac{\mu_{w}\left(T\right)}{T}=\mathrm{const}.\Rightarrow D_{H^{+},w}=D_{H^{+},w}^{ref}\frac{T}{T^{ref}}\frac{\mu_{w}^{ref}}{\mu_{w}\left(T\right)}\;$$
where $D_{H^{+},w}^{ref}$ and $\mu_{w}^{ref}$ are the diffusivity and dynamic viscosity of the fluid phase at the reference temperature (80 °C), respectively. $D_{H^{+},w}^{ref}\approx 9\times 10^{-5}{\;\mathrm{cm}}^{2}s^{-1}$ based on the works of Verbugge & Hill [37] and Ureña et al. [13], while $\mu_{w}\left(T\right)$ is given by the following correlation in the temperature range T=2−95 °C [38]
(25)$$\mu_{w}=10^{-3}\exp\left(-3.63+\frac{542.05}{T-144.15}\right)\;\;\;\;\;\;\left[\frac{\mathrm{kg}}{m\;s}\right]\;$$

The average proton concentration in the fluid phase, $C_{H^{+}}^{f}$, is determined by the electroneutrality condition
(26)$$\sigma A_{f}+FV_{f}C_{H^{+}}^{f}=0\Rightarrow \sigma a_{f}+FC_{H^{+}}^{f}=0\Rightarrow C_{H^{+}}^{f}=-\frac{\sigma a_{f}}{F}\;$$
where $A_{f}$ and $V_{f}$ are the wet area and volume, respectively, and $a_{f}=2I_{-1}\left(r_{a},r_{b}\right)$ is the specific surface area of hydrated cylindrical tubes (per unit of fluid volume). The average charge density, σ, is determined based on IEC, according to the following expression [39]
(27)$$\sigma =-\frac{FC_{H^{+},fh}^{f}}{a_{f,fh}};\;\;C_{H^{+},fh}^{f}=\frac{IEC\left(1-\phi_{v,fh}\right)\rho_{wet,fh}}{\phi_{v,fh}}\;$$

Combining Equations (26) and (27) yields
(28)$$C_{H^{+}}^{f}=C_{H^{+},fh}^{f}\frac{a_{f}}{a_{f,fh}}\;$$

In this expression, $C_{H^{+},fh}^{f}$, $\phi_{v,fh}$, $a_{f,fh}$ and $\rho_{wet,fh}$ are the average proton concentration in the fluid phase, water volume fraction, specific surface area and PEM density under fully humidified conditions (RH≈1). Thus, $C_{H^{+}}^{f}$ gradually increases from 0 under dry conditions to $C_{H^{+},fh}^{f}$ under fully humidified conditions [35].

### 4.4. Effective Proton Conductivity at the Cluster Network Scale

The above quantities at the cluster scale are transformed into volume-averaged quantities in a PEM introducing the effect of the volume ratio of the conductive phase and the connectivity of hydrated tubes. The effective proton conductivity at the cluster network scale can be written as
(29)$$\kappa^{eff}=\frac{\phi_{v}}{\tau }\kappa_{c}\;$$
where the water volume fraction, $\phi_{v}$, is given by Equation (17), and τ is the tortuosity factor. The percolative behavior of τ as a function of RH can be expressed as [13]
(30)$$\tau =\{\begin{array}{l} \infty \\ {\left(\frac{1-{\mathrm{RH}}_{th}}{\mathrm{RH}-{\mathrm{RH}}_{th}}\right)}^{n_{\tau }}\tau_{fh} \end{array}\;\;\;\;\;\begin{array}{c} \mathrm{if}\;\mathrm{RH}\leq {\mathrm{RH}}_{th}\;\\ \mathrm{if}\;\mathrm{RH}>{\mathrm{RH}}_{th} \end{array}\;$$
where ${\mathrm{RH}}_{th}$ is the percolation threshold, $\tau_{fh}$ is the tortuosity of the ionic network under fully humified conditions, and the exponent $n_{\tau }$ controls the increase of the tortuosity from $\tau =\tau_{fh}$ at RH=1 to τ=∞ when $\mathrm{RH}\leq {\mathrm{RH}}_{th}$.

## 5. Discussion of Results

The discussion of results is divided into two sections. Section 5.1 is devoted to the pore size distributions from PALS and ionic spacing estimated with the model. Section 5.2 is devoted to the experimental and numerical calculations of proton conductivity.

### 5.1. Pore Size Distribution and Ionic Spacing

Figure 5 shows the effect of RH% (0–80%) on the *o*-Ps lifetime and the free volume of the SPES PEMs, together with the corresponding pore size distributions. The *o*-Ps lifetime of the three PEMs can be arranged in ascending order as: SPES 1, SPES 2 and SPES 3. The pore size increases with IEC. A higher IEC increases the free volume and water uptake [40]. Two zones can be distinguished in the variation of the average pore size with RH%. SPES 2 and SPES 3 (DS=0.70−0.79) exhibit nearly identical behavior. At RH% equal to or lower than 30%, the *o*-Ps lifetime of the two PEMs decreases and therefore the pore size is reduced. When RH% is higher than 30%, the pore size of both PEMs increases. In contrast, SPES 1 (DS=0.45) with the lowest water uptake shifts the inflection point to 50% RH instead of 30% RH of the other two PEMs. The descent and ascent of the pore size with RH can be explained by the re-arrangement of the ionic network upon hydration. At low RH%, water molecules gradually fill the sulfonated pore space until they are almost evenly distributed in most pores. Below the inflection point, conformational changes take place in the ionic network before connected clusters are formed through the polymer membrane—that is, affinity of water molecules toward polar sulfonic groups against hydrophobic segments [31]. Increasing RH%, the PEM absorbs more water, and water molecules start to form ionic clusters in sulfonated domains. The amount of unbound water increases accompanied by a raise in the diffusion coefficient of water [41]. Once a (self-organized) connected network is formed, the water uptake is accompanied by an increase in pore size due to inflation of sulfonated pores, which outweighs the rearrangement of hydrophilic/hydrophobic segments. The amount of unbound water increases accompanied by a raise in the diffusion coefficient of water [41]. The increase of pore size and water uptake is particularly strong for SPES 3. This result agrees with the lack of separation of PSU and PPSU blocks at high IEC reported by Ureña et al. [12,13]. For instance, the mechanical properties of SPES PEMs significantly decreased at exceedingly high IEC due to excessive water uptake.

Figure 6 shows the temperature dependence of *o*-Ps lifetime, free volume and pore size distribution of the SPES PEMs. The free volume and the average pore size slightly increase with T due to the movement and dilation of polymer chains. Unlike the effect of RH, no significant changes in the pore size with T are found. This result confirms the stability of the SPES PEMs in the usual range of PEMFC operating temperatures. In fact, the SPES PEMs showed good mechanical properties in a wide temperature range in previous thermogravimetric analyses, with a glass transition temperature and a decomposition temperature of sulfonic groups around 200 °C [12].

Figure 7 shows the variation of the characteristic spacing between ionic clusters estimated with the model for the SPES PEMs (colored lines), together with the data determined for Nafion^®^ NRE-212 (black symbols) based on the free volume data, $V_{w}$, reported by Mohamed et al. [40] at different RHs and temperatures. The characteristic ionic spacing of Nafion^®^ NRE-212 was estimated as
(31)$$d_{c}\approx {\left(\frac{V_{w}}{\varepsilon_{w}}\right)}^{1/3}$$
where $\varepsilon_{w}\approx 0.28$ is the water-filled porosity [35].

The characteristic ionic spacing of the SPES PEMs is around 1.3 times lower than that of Nafion^®^ NRE-212 (0.7–0.75 nm vs. 0.85–1 nm) under vapor-equilibrated conditions. Both PEMs show a gradual increase of $d_{c}$ with RH due to swelling of the polymer matrix, increasing by $\Delta d_{c}∼0.05\;\mathrm{nm}$ in the range RH% = 0–80. However, the increase of $d_{c}$ with T of the SPES PEMs is significantly lower ($\Delta d_{c}∼0.01\;\mathrm{nm}$) compared to Nafion^®^ NRE-212 ($\Delta d_{c}∼0.15\;\mathrm{nm}$). This result agrees with the small effect of T on the pore size distribution observed with PALs and the good, stable mechanical properties of the SPES PEMs [13].

### 5.2. Water Volume Fraction and Proton Conductivity

The variation of the predicted and measured water volume fraction, $\phi_{v}$, as a function of RH and T is shown in Figure 8. Good agreement is found between the results. $\phi_{v}$ increases according to DS, with a higher increment between SPES 1 and SPES 2 (DS=0.45 vs. DS=0.70) compared to SPES 2 and SPES 3 (DS=0.70 vs. DS=0.79). The influence of RH on $\phi_{v}$ is significantly higher than that of T, ranging from 0 at RH% = 0 (assuming no residual water) up to $\phi_{v}=20\%$ at RH% = 80 for SPES 3. In contrast, the variation of $\phi_{v}$ in the temperature range T=30−90 °C is lower than 5%. The WU of the SPES PEMs is comparable to that of Nafion^®^ 212 at high RH ($\phi_{v}=20-30\%$ at RH% = 80–100).

Figure 9 shows the numerical and experimental effective ionic conductivity, $\kappa^{eff}$, as a function of RH and T, together with previous experimental data reported for Nafion^®^ 212 [42,43,44,45]. The calculated proton concentrations and surface charge densities are provided in Appendix A. $\kappa^{eff}$ increases with IEC and DS in the following order: SPES 3 > SPES 2 > SPES 1. The growth of $\kappa^{eff}$ is stronger between SPES 1 and SPES 2 owing to the nonlinear coupling between DS and τ (see [13] for further details). In all cases, the influence of RH and T is similar. $\kappa^{eff}$ increases with RH according to a percolation law of the form ${\left(\mathrm{RH}-{\mathrm{RH}}^{th}\right)}^{n}$ due to the increase of the number of hydrated sites (i.e., decrease of the cluster-network tortuosity). In addition, $\kappa^{eff}$ increases rather linearly with T mainly due to the increase of the proton mobility and to a lesser extent due to the slight growth of the average size of ionic clusters.

The ionic conductivity of SPES 2 and 3 is around two times lower than that of Nafion^®^ 212. The reduced proton conduction in the copolymer PEMs can be explained by the lower radius of ionic clusters ($r^{avg}\approx 0.25\;\mathrm{nm}$ vs. $r^{avg}\approx 0.35\;\mathrm{nm}$, $V_{h}\approx 0.09\;{\mathrm{nm}}^{3}$ vs. $V_{h}\approx 0.23\;{\mathrm{nm}}^{3}$ at RH%≈80 and T≈30 °C [31,40]) at similar proton concentration, $C_{H^{+},fh}^{f}$, and water volume fraction, $\phi_{v}$. A lower free volume fraction (pore radius and ionic spacing) decreases proton conductivity at the cluster scale due to a reduction of the bulk space where proton transfer is faster (i.e., the activation free energy is lower compared to the surface region) [46,47]. In addition, a lower free volume fraction typically increases the tortuosity of the ionic network at the cluster-network scale, as predicted by Kozeny-Carman’s theory [35,48]
(32)$$\phi_{v}∼n_{A}\pi r^{2}\tau \Rightarrow \tau ∼\frac{\phi_{v}}{n_{A}\pi r^{2}}$$
where $n_{A}$ is the number of ionic tubes per unit of geometric area. For instance, Rao et al. [49] reported ionic conductivities as high as $400\;{\mathrm{mS}\;\mathrm{cm}}^{-1}$ in water-equilibrated Nafion^®^ 117 (δ≈180 μm) treated with ultraviolet radiation due to an increase of the water volume fraction from 0.28 (pristine Nafion^®^) to 0.40 (optimized Nafion^®^), while keeping moderate water uptake (WU≈23.5%) and swelling ratio (SW≈19.2%) and similar methanol permeability to the pristine sample.

## 6. Conclusions

The free volume and the ionic conductivity of proton exchange membranes (PEMs) based on multiblock copolymers of sulfonated polysulfone (SPSU) and polyphenylsulfone (SPPSU) with different degrees of sulfonation have been examined experimentally as a function of relative humidity (RH% = 0–80) and temperature (T=30−80 °C). Free volume was characterized using positron annihilation lifetime spectroscopy (PALS) and ionic conductivity using electrochemical impedance spectroscopy (EIS). Experimental observations were compared with the predictions of a bundle-of-tubes model.

The free volume of the copolymer PEMs varied slightly with temperature in agreement with their good thermo-mechanical properties. However, the free volume (pore radius and ionic spacing) of the copolymer PEMs was significantly lower than that of Nafion^®^ 212 despite their comparable water uptake. The lower free volume of the SPES PEMs explains their lower ionic conductivity compared to Nafion^®^ due to the decrease of the bulk volume available for proton transport and the increase of the tortuosity at the cluster-network scale.

Future work shall focus on the optimization of the length of copolymer blocks to increase ionic conductivity, while keeping good chemical and mechanical stability. In addition, nanoparticles can be added to increase ion exchange capacity and hydrophilicity. The interplay between block length, polymer density and degree of sulfonation on pore size-tortuosity, ionic mobility and conductivity should be examined both experimentally and numerically for optimization of ionic conductivity. Specifically, an optimal design of the block length of SPSU/SPPSU membranes is expected to provide a better micro-separation of hydrophilic/hydrophobic domains.

## Figures and Tables

**Figure 1 polymers-14-01688-f001:**
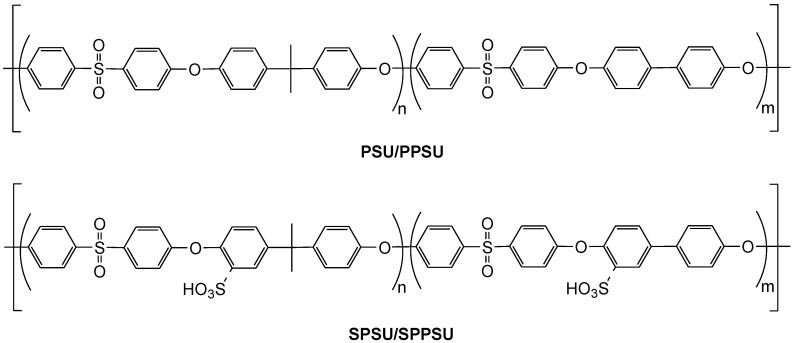
Chemical structure of polysulfone/polyphenylsulfone (PSU/PPSU) and sulfonated polysulfone/polyphenylsulfone (SPSU/SPPSU) copolymers (reproduced from [13]).

**Figure 2 polymers-14-01688-f002:**
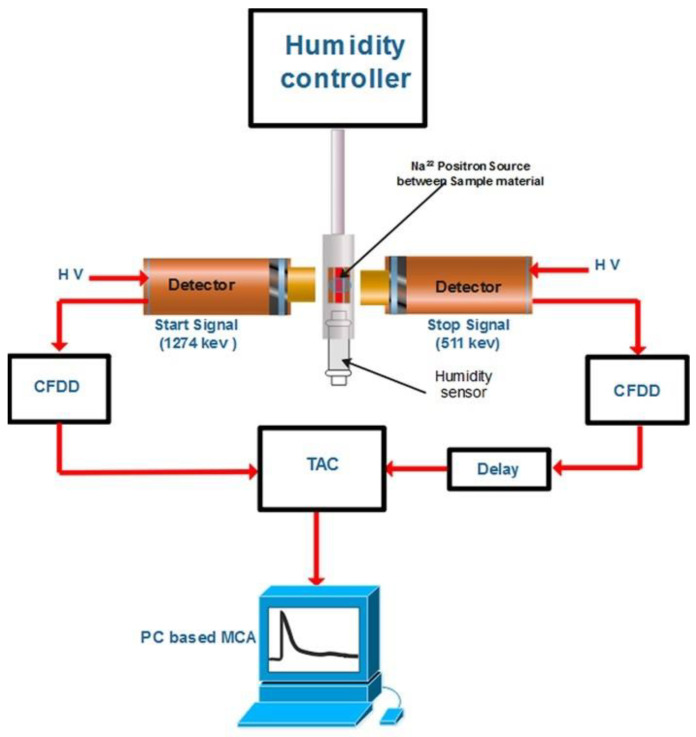
Setup of the PALS system. The main components of the system are the constant fraction differential (CFDD), the discriminator (CFDD), the time-to-amplitude converter (TAC) and the multichannel analyzer (MCA).

**Figure 3 polymers-14-01688-f003:**
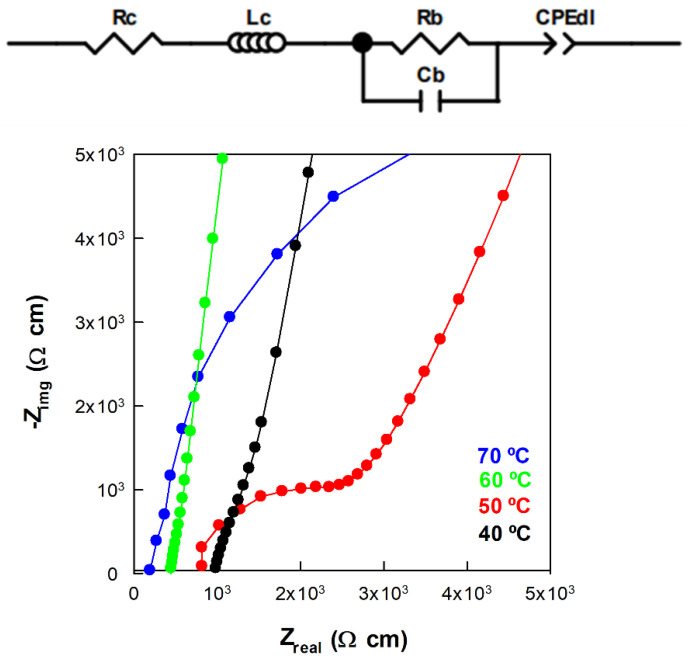
Equivalent circuit associated with the SPES membranes sandwiched between two electrodes. Nyquist plots of SPES 2 at different temperatures. Some curves are not included for clarity.

**Figure 4 polymers-14-01688-f004:**
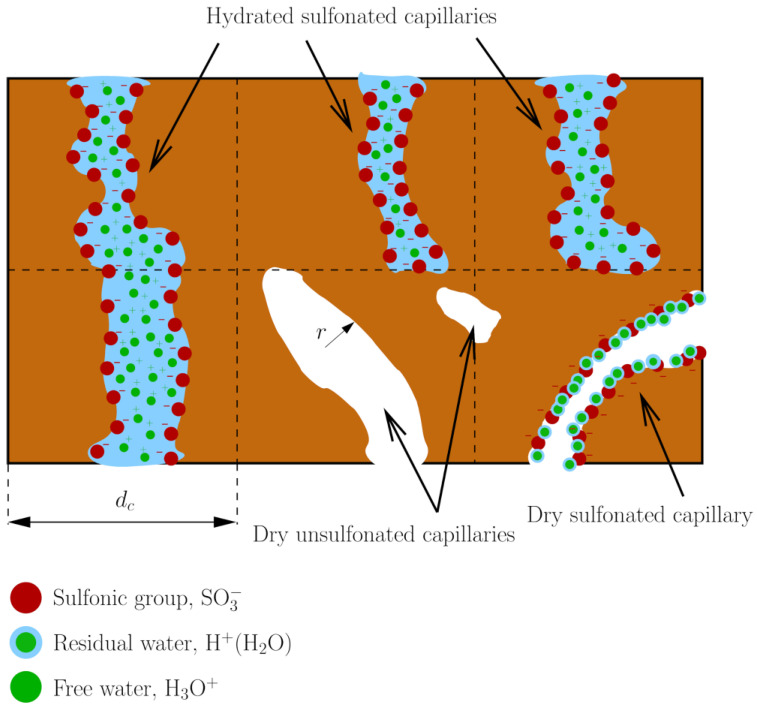
Schematic of the bundle-of-tubes model used to analyze proton conductivity of copolymer PEMs. Three types of pore bodies are considered: hydrated sulfonated sites, dry sulfonated sites, and dry non-sulfonated sites. The radius r of the tubes is distributed according to the pore size distribution, PSD(r), determined from PALS. The average distance between tubes is $d_{c}$.

**Figure 5 polymers-14-01688-f005:**
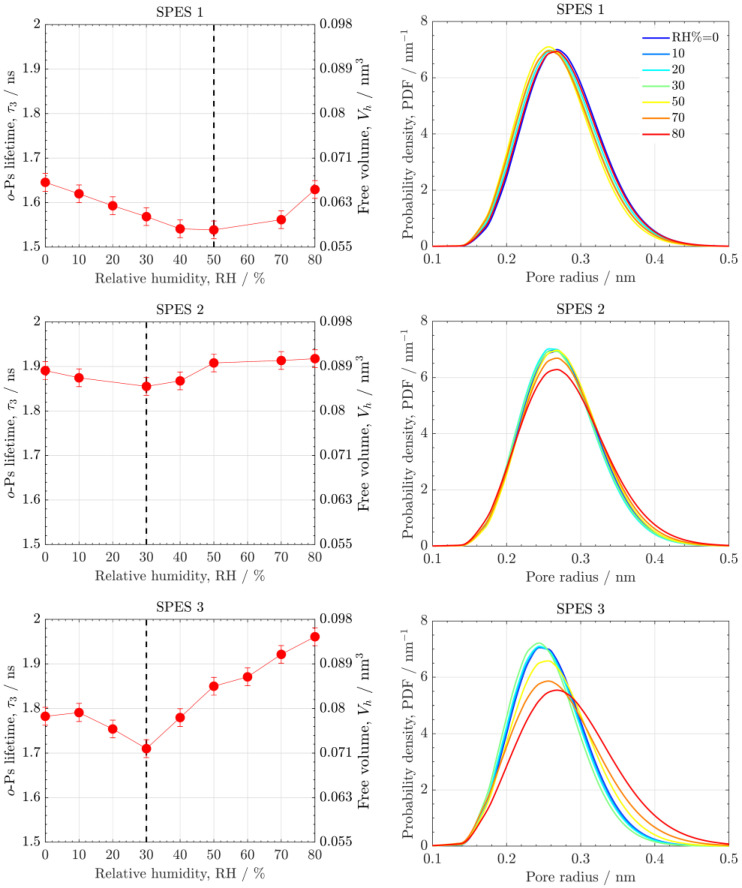
(**left**) *o*-Ps lifetime, $\tau_{3}$, and free volume, $V_{h}$, and (**right**) pore size distributions as a function of pore radius of SPES PEMs at different relative humidities (RH% = 0–80). The inflection point of RH is indicated by a dashed line.

**Figure 6 polymers-14-01688-f006:**
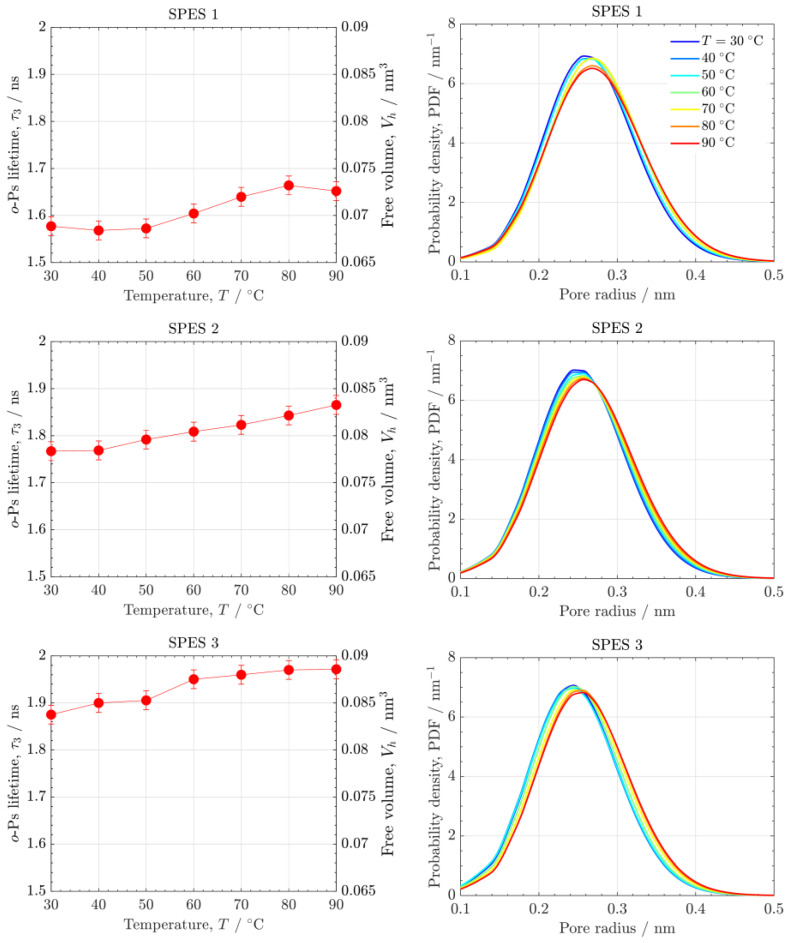
(**left**) *o*-Ps lifetime, $\tau_{3}$, and free volume, $V_{h}$, and (**right**) pore size distributions as a function of pore radius of the SPES PEMs at different temperatures (T=30−90 °C ).

**Figure 7 polymers-14-01688-f007:**
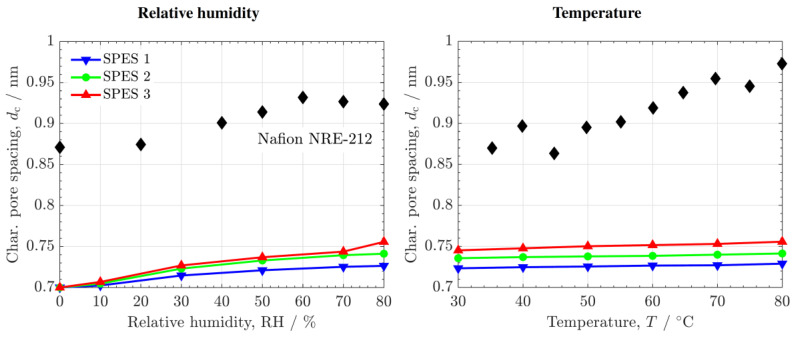
Characteristic ionic or pore spacing, $d_{c}$, estimated with the bundle-of-tubes model as a function of relative humidity, RH, and temperature, T. The data corresponding to Nafion^®^ NRE-212 (black symbols) are included for comparison.

**Figure 8 polymers-14-01688-f008:**
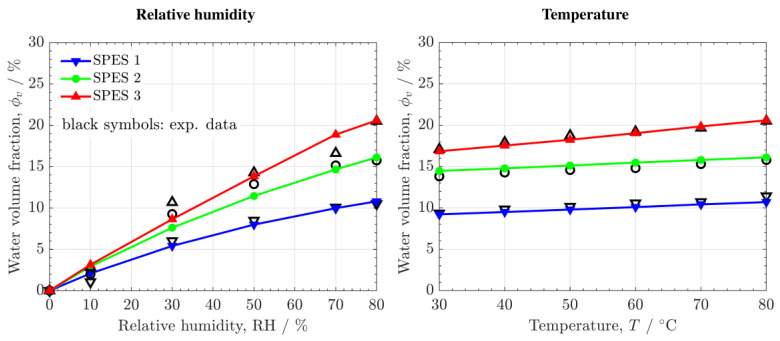
Numerical (colored lines) and experimental (open symbols) water volume fraction, $\phi_{v}$, as a function of relative humidity, RH, and temperature, T.

**Figure 9 polymers-14-01688-f009:**
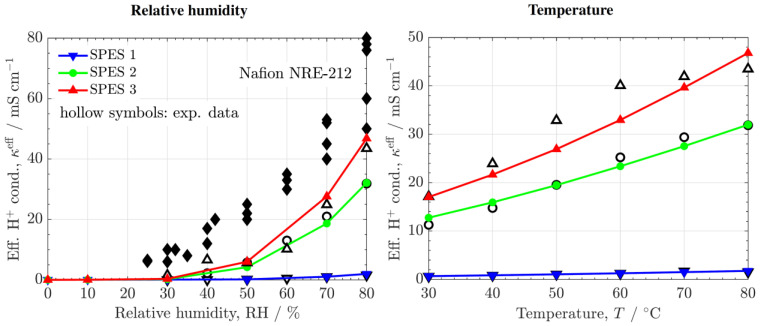
Numerical (colored lines) and experimental (open symbols) effective ionic conductivity, $\kappa^{eff}$, as a function of relative humidity, RH, and temperature, T. The experimental data corresponding to Nafion^®^ 212 (black symbols) are included for comparison.

**Table 1 polymers-14-01688-t001:** Ion exchange capacity (IEC), degree of sulfonation (DS) and dry density ($\rho_{dry}$) of the SPES membranes [12] and Nafion^®^ NRE-212 [11].

	SPES 1	SPES 2	SPES 3	Nafion^®^ NRE-212
IEC/${\mathrm{meq}\;g}^{-1}$	0.97	1.46	1.62	0.91
DS/−	0.45	0.70	0.79	1
$\rho_{dry}\;$/${\mathrm{kg}\;m}^{-3}$	1140	1140	1140	1.97

**Table 2 polymers-14-01688-t002:** Water uptake, WU%, and ionic conductivity, κ, of the SPES PEMs at T=80 °C (RH%=0−80) and RH%=80 (T=30−80 °C).

		SPES 1		SPES 2		SPES 3	
T	RH	WU	κ	WU	κ	WU	κ
°C	%	%	$mS\;cm^{-1}$	%	$mS\;cm^{-1}$	%	$mS\;cm^{-1}$
80	0	0	-	0	-	0	-
80	10	0.91	-	1.74	-	2.51	-
80	20	-	-	-	-	-	-
80	30	5.46	0.01	8.71	0.60	10.21	1.53
80	40	-	0.03	-	2.19	-	6.60
80	50	7.92	0.18	12.61	5.73	14.22	5.64
80	60	-	0.51	-	13.02	-	10.20
80	70	9.55	1.02	15.22	20.97	17.02	24.90
80	80	10.02	1.53	15.95	31.86	22.01	43.5
30	80	9.01	0.61	14.02	11.28	18.03	17.11
40	80	9.51	0.82	14.52	14.77	19.02	23.92
50	80	9.82	1.06	14.81	19.54	20.05	32.85
60	80	10.21	1.23	15.03	25.22	20.54	40.08
70	80	10.31	1.69	15.52	29.39	21.01	41.92

**Table 3 polymers-14-01688-t003:** Reference values used for bilinear interpolation of the characteristic radius, $r_{c}$, and standard deviation, σ, and parameters used in the bundle-of-tubes model.

	SPES 1	SPES 2	SPES 3
Ref. char. radius, $r_{c,11},r_{c,12},r_{c,21},\;r_{c,22}\;/\mathrm{nm}$	0.270, 0.275, 0.271, 0.290	0.263, 0.265, 0.273, 0.287	0.251, 0.260, 0.276, 0.303
Ref. std. dev., $\sigma_{11},\;\sigma_{12},\;\sigma_{21},\;\sigma_{22}\;/\mathrm{nm}$	0.207, 0.220, 0.204, 0.214	0.211, 0.222, 0.223, 0.235	0.218, 0.222, 0.250, 0.262
Humidification coefficients,$\;a_{1},a_{2},a_{3}/-$	0.3, −1.25, 1.95		
Fully hum. tortuosity, $\tau_{fh}/-$	35	3.4	2.4
RH threshold, ${\mathrm{RH}}_{th}/-$	0.3	0.19	0.18
Tortuosity exponent, $n_{\tau }/-$	2.5		
Dry char. spacing, $d_{o}/\mathrm{nm}$	0.7		

## Data Availability

Not applicable.

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
