# Peer review of "Characterization and Modeling of Free Volume and Ionic Conduction in Multiblock Copolymer Proton Exchange Membranes"

_polymers, 2022, doi:10.3390/polym14091688_

Round 1

Reviewer 1 Report

The authors present an interesting model study by investigating the relationship between free volume and ionic conductivity of SPES PEMS. The study provides insights into transport mechanism that is important to the performance of the PEMFC. However, the following points need to be addressed before the acceptance of the paper for publication in Polymers.

  1. The author needs to give a thorough introduction about the motivation of studying sulfonated PSU and PPSU-based block copolymer. What is the advantage of using “hydrophilic-block-hydrophilic” copolymer compared to “hydrophobic-block-hydrophilic” copolymer? For example, what is the benefit of using SPSU-SPPSU-based block copolymer compared to PS(polystyrene) -SPSU block copolymer?
  2. On page 3, equation (1): the numerator should have an addition sign (not a subtraction sign)
  3. The author must show the Nyquist plots from EIS measurements, and the equivalent circuit used to extract the film resistance.
  4. For Figure 4, the author needs to explain the phenomenon that the pore size decreases as the increase of RH when RH is less than 30%. Even at low RH, water molecules will gradually fill the sulfonated pores, which should still inflate the sulfonated pores.
  5. What is the glass transition temperatures (Tg) for SPES1, SPES2, and SPES3? Is it possible to measure Tg at different relative humidity? In figure 5, the free volume of SPES PEMS remains almost unchanged from 30 °C to 90 °C might be attributed to the high Tg of the polymer.
  6. In equation 29, what is the definition of κc? For Figure 8, the author needs to elaborate the reason of plotting effective conductivity (κeff) versus RH rather than conductivity (κ) versus RH.
  7. On page 15, the thickness of Nafion 117 should be ca. 180 µm instead of 10 µm.

Author Response

The authors present an interesting model study by investigating the relationship between free volume and ionic conductivity of SPES PEMS. The study provides insights into transport mechanism that is important to the performance of the PEMFC. However, the following points need to be addressed before the acceptance of the paper for publication in Polymers.

  1. The author needs to give a thorough introduction about the motivation of studying sulfonated PSU and PPSU-based block copolymer. What is the advantage of using “hydrophilic-block-hydrophilic” copolymer compared to “hydrophobic-block-hydrophilic” copolymer? For example, what is the benefit of using SPSU-SPPSU-based block copolymer compared to PS(polystyrene) -SPSU block copolymer?

The main motivation of using “hydrophilic-block-hydrophilic” copolymer is to increase charge concentration and thus ionic mobility, while keeping good mechanical properties. The use of “hydrophobic-block-hydrophilic” copolymer by concentrating charge in one block and using the other one as the backbone of the set is also an interesting approach. However, the tailored synthesis and design of the former option is easier from an industrial point of view. In addition, PSU/PPSU blocks are compatible and allow the preparation of copolymers with high molecular weight.

Future work will focus on determining optimum block length (and pore size) for maximizing ionic conductivity and durability of PSU and PPSU-based block copolymer.

The above aspects have been incorporated into the introduction.

  1. On page 3, equation (1): the numerator should have an addition sign (not a subtraction sign)

This typo has been corrected.

  1. The author must show the Nyquist plots from EIS measurements, and the equivalent circuit used to extract the film resistance.

As an example, the Nyquist plots of SPSU and SPPSU membranes are shown in Figure 3 of the new version of the manuscript, as well as the equivalent circuit.

  1. For Figure 4, the author needs to explain the phenomenon that the pore size decreases as the increase of RH when RH is less than 30%. Even at low RH, water molecules will gradually fill the sulfonated pores, which should still inflate the sulfonated pores.

This phenomenon is explained by conformational changes in the ionic network before connected clusters are formed through the polymer membrane. That is, affinity of water molecules toward polar sulfonic groups against hydrophobic polymer segments. Once a (self-organized) connected network is formed, water uptake is accompanied by an increase of pore size due to inflation of sulfonated pores, which outweighs rearrangement of hydrophilic/hydrophobic copolymer segments. Note the correlation between the increase of the ionic conductivity and the inflection point of the pore size of the SPES membranes. In addition, the diameter of a water molecule is about 0.276 nm (the volume is about 0.011 nm3), which is less than the diameter of the free volume holes in the SPES membranes, water molecules find a suitable site with enough size to occupy under low RH conditions. As a result, water begins to fill membrane's intermolecular gaps. This reduces the pickoff annihilation, resulting in a minor reduction in the free volume hole.

This explanation has been incorporated into Section 5.1 for the sake of clarity.

  1. What is the glass transition temperatures (Tg) for SPES1, SPES2, and SPES3? Is it possible to measure Tgat different relative humidity? In figure 5, the free volume of SPES PEMS remains almost unchanged from 30 °C to 90 °C might be attributed to the high Tg of the polymer.

The glass transition temperature of the dry SPES membranes is between 180-190 oC, slightly lower than the decomposition temperature of sulfonic groups (240-350 oC) [1]. Both temperatures are significantly higher than the examined range between 30-90 oC in which the free volume remains rather constant.

The dependence of the glass transition temperature on RH of the SPES membranes could be measured using a climate chamber. However, no significant changes are expected so as to affect the pore size distribution in the range of interest between 30-90 oC as found in the free volume measurements. For this reason, experimental results of glass transition temperature vs. RH have been omitted in the revised version of the manuscript.

[1] N. Ureña, M.T. Pérez-Prior, C. del Río, A. Várez, J.-Y. Sanchez, C. Iojoiu, B. Levenfeld, Multiblock copolymers of sulfonated PSU/PPSU Poly(ether sulfone)s as solid electrolytes for proton exchange membrane fuel cells, Electrochimica Acta. 302 (2019) 428–440. https://doi.org/10.1016/j.electacta.2019.01.112.

  1. In equation 29, what is the definition of κc? For Figure 8, the author needs to elaborate the reason of plotting effective conductivity (κeff) versus RH rather than conductivity (κ) versus RH.

κc is the (effective) ionic conductivity at the cluster scale (i.e., within hydrated tubes), which is determined as described in Section 4.3. The word “effective” has been removed from the title of Section 4.3 for clarity.

κeff is the effective ionic conductivity at the cluster-network scale across the membrane, which is determined as described in Section 4.4. This conductivity includes the effect of water volume fraction and tortuosity of the ionic network, and it is the one of interest since it can be compared with experimental data.

  1. On page 15, the thickness of Nafion 117 should be 180 µm instead of 10 µm.

This typo has been corrected.

Reviewer 2 Report

In this manuscript, PALS and EIS techniques are used to characterize the pore size distribution and ionic conductivity of synthesized PEMs with different degrees of sulfonation. The results show that the free pore size changes little with temperature, while the free volume (pore size and ionic spacing) is significantly lower than perfluorosulfonic acid PEMs. It can be concluded that nano-structured copolymer PEMs with high degree of sulfonation and charge concentration are promising candidates for durable, high-performance proton exchange electrochemical devices.

I consider the content of this manuscript will definitely meet the reading interests of the readers of the Polymers journal. Therefore, I suggest giving a minor revision and the authors need to clarify some issues or supply some more data to enrich the content.

  1. Abstract and Introduction

  • For the Keywords, ‘electrochemical impedance spectroscopy’, ‘SPES’, and ‘charge concentration’ should also be added to attract a broader readership and highlight the significance of this work.

  • Please pay attention to grammar and spelling problems, especially the missing or redundant definite articles. I suggest double-checking the whole manuscript. I will point out several examples, but unfortunately, I cannot point out all of them. For example:

Line 111, ‘The synthesized PES copolymer was dissolved in dry 1,2-dichloroethane (DCE)

under an inert atmosphere at ambient temperature’;

Line 131, ‘The ionomers were dissolved in DMAc (5 wt%) and cast onto a Petri glass and

dried under vacuum for over 48 h’;

Line 217, ‘The relationship of ... on RH and T extracted from PALS’;

Line 264, ‘The whole range of tube radii is hydrated when the pore space of the PEM is fully sulfonated (DS=1) and filled with water. ’and so on.

  • Line 29, ‘Proton exchange membrane fuel cells (PEMFCs) are receiving considerable attention for small stationary applications and the automotive industry because of their attractiveness as efficient and clean energy converters [1].

    For stationary applications, what are the benefits of PEMFCs compared to the cost-effective redox flow batteries that do not require Pt-based catalysts [Journal of Power Sources 493 (2021): 229445]? For automotive industry, lithium batteries are more commonly used than PEMFCs currently [Renewable and Sustainable Energy Reviews 78 (2017): 834-854.]. What is the potential of PEMFCs in the field of electric vehicles? Here, a brief comparison between PEMFCs and the other two technologies needs to be supplemented. Otherwise, does it really make sense to develop and design membranes for PEMFC systems with expensive catalysts?

  • Line 34 to 38, ‘This ionomer exhibits many interesting properties, including good chemical stability and high proton conductivity, as well as high water uptake...

    It is best to explain the properties in combination with the chemical structure of the Nafion membrane. Nafion is composed of hydrophobic PTFE backbones (provide good chemical stability) and hydrophilic sulfonic acid groups (high water uptake). When hydrated, there are significant phase separations between the hydrophobic and hydrophilic domains, which provide the channels for proton transportation [Solid State Ionics 319 (2018): 110-116]. Due to the special structure, the complex production procedures leads to the high cost of Nafion.

  • Line 53, ‘Multiple PEM characteristics that affect cell performance... are all strongly correlated with the structure of the PEM free volume holes.

    What are the free volume holes? This issue should be further explained. For example, ‘It is known that the mechanical and rheological properties of a polymer are influenced by molecular motion or chain flexibility of the polymer molecules, which in turn depends upon the microscopic free-volume holes, present mainly in the amorphous region in the polymer. It is widely accepted in view of its ability to predict many properties of polymers at molecular level. [Journal of Polymer Science Part B: Polymer Physics 36.6 (1998): 983-989]’

  • Line 76, ‘These multiblock copolymers were chosen because they could be manufactured on a large scale, (ii) they had good water absorption and mechanical strength...There should be first (i), and then (ii).

  1. Materials and Modeling

  • In Equation 1, why it is ‘nDSPSU-mDSPPSU’? It should be ‘+’.

  • In Table 1, the information of Nafion should also be added for comparison.

  • Three types of pore bodiesshould appear only once, for Line 211 and the caption of Figure 3, the identical information appears twice.

  • In this paper, PEM with the highest sulfonation degree is 2-3 orders of magnitude lower than Nafion in terms of ion conductivity. How will future work improve ionic conductivity? Combined with the focus of this paper, we should make corresponding prospects.

Author Response

In this manuscript, PALS and EIS techniques are used to characterize the pore size distribution and ionic conductivity of synthesized PEMs with different degrees of sulfonation. The results show that the free pore size changes little with temperature, while the free volume (pore size and ionic spacing) is significantly lower than perfluorosulfonic acid PEMs. It can be concluded that nano-structured copolymer PEMs with high degree of sulfonation and charge concentration are promising candidates for durable, high-performance proton exchange electrochemical devices.

I consider the content of this manuscript will definitely meet the reading interests of the readers of the Polymers journal. Therefore, I suggest giving a minor revision and the authors need to clarify some issues or supply some more data to enrich the content.

Abstract and Introduction

For the Keywords, ‘electrochemical impedance spectroscopy’, ‘SPES’, and ‘charge concentration’ should also be added to attract a broader readership and highlight the significance of this work.

The suggested keywords have been added.

Please pay attention to grammar and spelling problems, especially the missing or redundant definite articles. I suggest double-checking the whole manuscript. I will point out several examples, but unfortunately, I cannot point out all of them. For example:

Line 111, ‘The synthesized PES copolymer was dissolved in dry 1,2-dichloroethane (DCE)

under an inert atmosphere at ambient temperature’;

Line 131, ‘The ionomers were dissolved in DMAc (5 wt%) and cast onto a Petri glass and

dried under vacuum for over 48 h’;

Line 217, ‘The relationship of ... on RH and T extracted from PALS’;

Line 264, ‘The whole range of tube radii is hydrated when the pore space of the PEM is fully sulfonated (DS=1) and filled with water. ’and so on.

The writing has been revised and English improved.

Line 29, ‘Proton exchange membrane fuel cells (PEMFCs) are receiving considerable attention for small stationary applications and the automotive industry because of their attractiveness as efficient and clean energy converters [1].’

    For stationary applications, what are the benefits of PEMFCs compared to the cost-effective redox flow batteries that do not require Pt-based catalysts [Journal of Power Sources 493 (2021): 229445]? For automotive industry, lithium batteries are more commonly used than PEMFCs currently [Renewable and Sustainable Energy Reviews 78 (2017): 834-854.]. What is the potential of PEMFCs in the field of electric vehicles? Here, a brief comparison between PEMFCs and the other two technologies needs to be supplemented. Otherwise, does it really make sense to develop and design membranes for PEMFC systems with expensive catalysts?

PEMFC is a clean technology, which is having an increasing applicability for portable devices, e.g., unmanned aerial vehicles (UAVs), transportation applications (heavy duty tracks, light duty vehicles, trains, submarines, etc.) and forklifts, among other devices. The main advantage of PEMFCs compared to other electrochemical devices is the increased operational time required in the application, such as extended flight time of UAVs or extended range of good trucks, in combination with no emission of air pollutants, such as forklifts in enclosed facilities and widespread use of light duty vehicles.

The design of materials (e.g., membranes) with improved properties at low cost is an inherent need in electrochemical devices, which is continuously evolving, as it is the case of PEMFCs. As a matter of fact, the consequences of the war in Ukraine regarding lithium extraction are currently unknown and may have an important impact on European plans in terms of large-scale production of li-ion batteries.

This aspect has been clarified in the introduction.

Line 34 to 38, ‘This ionomer exhibits many interesting properties, including good chemical stability and high proton conductivity, as well as high water uptake...’

    It is best to explain the properties in combination with the chemical structure of the Nafion membrane. Nafion is composed of hydrophobic PTFE backbones (provide good chemical stability) and hydrophilic sulfonic acid groups (high water uptake). When hydrated, there are significant phase separations between the hydrophobic and hydrophilic domains, which provide the channels for proton transportation [Solid State Ionics 319 (2018): 110-116]. Due to the special structure, the complex production procedures leads to the high cost of Nafion.

The suggested change has been included.

Line 53, ‘Multiple PEM characteristics that affect cell performance... are all strongly correlated with the structure of the PEM free volume holes.’

    What are the free volume holes? This issue should be further explained. For example, ‘It is known that the mechanical and rheological properties of a polymer are influenced by molecular motion or chain flexibility of the polymer molecules, which in turn depends upon the microscopic free-volume holes, present mainly in the amorphous region in the polymer. It is widely accepted in view of its ability to predict many properties of polymers at molecular level. [Journal of Polymer Science Part B: Polymer Physics 36.6 (1998): 983-989]’

The gaps between entangled polymer chains provide free volume, which is an intrinsic characteristic of the polymer matrix [1]. According to the free volume model, the absorption and diffusion of molecules in polymers are highly dependent on the available free volumes. Many polymers, for example, show an increase in sorption as the number of free volumes increases.

This aspect has been clarified in the introduction.

[1] https://www.sciencedirect.com/topics/materials-science/free-volume

Line 76, ‘These multiblock copolymers were chosen because they could be manufactured on a large scale, (ii) they had good water absorption and mechanical strength...’There should be first (i), and then (ii).

This typo has been corrected.

Materials and Modeling

In Equation 1, why it is ‘nDSPSU-mDSPPSU’? It should be ‘+’.

This typo has been corrected.

In Table 1, the information of Nafion should also be added for comparison.

The information has been added to Table 1 according to [1].

[1] A. Kusoglu, A.Z. Weber, New Insights into Perfluorinated Sulfonic-Acid Ionomers, Chem. Rev. 117 (2017) 987–1104.

‘Three types of pore bodies’ should appear only once, for Line 211 and the caption of Figure 3, the identical information appears twice.

This typo has been corrected.

In this paper, PEM with the highest sulfonation degree is 2-3 orders of magnitude lower than Nafion in terms of ion conductivity. How will future work improve ionic conductivity? Combined with the focus of this paper, we should make corresponding prospects.

We disagree with the statement that the ionic conductivity is 2-3 orders of magnitude lower. The ionic conductivity of the SPES membranes is a factor 2-3 lower than Nafion.

Future work shall focus on the optimization of the length of copolymer blocks to increase ionic conductivity, while keeping good chemical and mechanical stability. In addition, nanoparticles can be added to increase IEC and hydrophilicity. The interplay between block length, polymer density and degree of sulfonation on pore size-tortuosity, ionic mobility and conductivity should be examined both experimentally and numerically for optimization. This aspect has been included in the conclusions.